

# A review of methods and software for polygenic risk score analysis

Sara Benoumhani[1], Areej Al-Wabil[1,2], Niddal Imam[3],
Bashayer Alfawaz[1], Amaan Zubairi[1], Dalal Aldossary[1] and
Mariam AlEissa[1,4,5,6,7]

[1] Artificial Intelligence Research Center, Alfaisal University, Riyadh, Saudi Arabia
[2] Software Engineering Department, Alfaisal University, Riyadh, Saudi Arabia
[3] College of Computing and Informatics, Saudi Electronic University, Riyadh, Saudi Arabia
[4] Molecular Genetics Laboratory, Public Health Authority, Riyadh, Saudi Arabia
[5] College of Medicine, Alfaisal University, Riyadh, Saudi Arabia
[6] King Khaled Eye Specialist Hospital (KKESH) Research Centre, Riyadh, Saudi Arabia
[7] Computational Sciences Department at the Centre for Genomic Medicine (CGM), King Faisal
Specialist Hospital and Research Center, Riyadh, Saudi Arabia

## ABSTRACT

Polygenic risk scores (PRSs) are emerging as powerful tools for predicting individual susceptibility to various diseases and traits based on genetic variants. These scores integrate information from multiple genetic markers associated with the trait or disease of interest, offering personalized risk assessment and enhancing disease management strategies. PRS is an active area of research and is being studied in various fields, such as disease prediction. This review explores the advancement of PRS research, focusing on methodological approaches, software tools, and applications across diverse disciplines. A systematic literature review identified 40 relevant articles classified based on PRS methods and software. Key methods for PRS computation, including penalized regression and threshold-based approaches, Bayesian approaches, and machine learning approaches, are discussed, along with notable software and their features. Applications of PRS in disease prevention are highlighted. Challenges and future directions, such as increasing diversity in genetic data, integrating environmental factors, and evaluating clinical implications, are also discussed to guide future research and implementation efforts.

## INTRODUCTION

The ability to predict complicated traits and illnesses, such as cancer from an individual's genetic variations is important for effective illness prevention (*Wray et al., 2013*; *Chatterjee, Shi & García-Closas, 2016*; *Yang et al., 2017*; *Muñoz et al., 2016*; *Wang et al., 2017*; *Visscher et al., 2017*). Polygenic risk scores (PRSs) are calculated by the effect sizes of multiple genetic variants known to be associated with the disease or trait of interest. Researchers are improving risk prediction for common diseases using genetic data. Risk scores that incorporate both clinical risk indicators and PRSs for a specific illness would significantly improve the accuracy of lifetime risk prediction and the intuitiveness of

Corresponding author
Mariam AlEissa,
maaleissa@alfaisal.edu

disease risk management (*Slunecka et al., 2021*). Additionally, PRS's research has been expanded to include many diseases using many methods, such as machine learning (ML) today. The PRS has the potential to predict individual disease risks and potentially offer a more effective predictor with improved discrimination properties compared to one based solely on established markers (*Dudbridge, 2013*). Over the last fifteen years, the escalating presence of PRS research groups, the proliferation of peer-reviewed journals, and the surge in conference abstracts all serve as indicators of the rapidly expanding interest in this field. Researchers have been investigating PRS to understand disease risk, predict outcomes, and potentially inform clinical decisions. These studies received a high number of citations in recorded time (*Dudbridge, 2013*; *Lewis & Vassos, 2020*; *Mavaddat et al., 2019*). Moreover, several companies have joined forces with research groups to advance PRS-related technologies, delineating clear roadmaps for their development (*Slunecka et al., 2021*). This remarkable growth in PRS's research is closely tied to an influx of researchers from diverse disciplines, which fostering an interdisciplinary approach that has led to the creation of PRS systems tailored for various target applications. Since 2018, there has been a growing interest in using PRSs to predict the risk of developing multiple diseases; numerous research studies have demonstrated that PRSs are capable of predicting disease status (*Mavaddat et al., 2019*; *Wray et al., 2018*; *Khera et al., 2018*). Researchers have also been exploring ways to improve the accuracy of PRSs by incorporating additional data, such as environmental factors (*Musliner et al., 2019*; *Lewis & Hagenaars, 2019*). PRSs have been used in various applications such as predicting disease risk (*Haas et al., 2018*), patient stratification (*Mavaddat et al., 2019*), investigating treatment response (*Shi et al., 2020*; *Mega et al., 2015*; *Natarajan et al., 2017*) and experimental perturbation informed by genetics (*Dobrindt et al., 2021*; *Hoekstra et al., 2017*). Most prominent PRS techniques, including those integrating functional annotation (*Márquez-Luna et al., 2021*; *Hu et al., 2017*), are based on the classical polygenic disease model. Recently, there has been a marked rise in the volume of studies, investigations, and articles centered on PRS tools. The diversity in research methodologies employed across these studies has yielded a wide spectrum of outcomes, influenced by numerous variables, such as the dataset's methods of calculation of the PRSs.

This study aims to grasp PRS software trends and examine previous studies to equip researchers with knowledge for forthcoming PRS software advancements. In this review, we found that PRS publications span a range of fields such as genetics, epidemiology, computer science, biostatistics, and mathematics. This diversity presents a challenge for comparative analysis due to the wide array of research focuses and methodologies across different journals and scientific areas.

The primary goal of this review is to evaluate the methods, including tools and software. We aim to establish a conceptual structure for the categorization of PRS-related studies, which will aid in the systematic review of PRS research literature. The subsequent sections will detail the proposed framework for categorizing PRS research. Initially, we will define the research approach. Subsequently, we will expound on the suggested categorization

framework for PRS research reviews. The findings are presented, offering insights for forthcoming research and deliberate the trends and challenges in PRS prediction tools. We conclude by summarizing the review's contributions to the body of knowledge in PRS prediction.

## METHODS AND MATERIALS

We conducted a systematic review of techniques by the PRISMA guidelines. The subsequent sections detail the methods for article extraction, including the criteria for article selection and the filtering methodologies employed.

### Data sources and procedures for the extraction of articles

Articles concerning PRS can be found dispersed throughout various academic journals that span multiple disciplines. We performed an initial search using online databases such as Web of Science, PubMed, Google Scholar, and Scopus. We used the Publish or Perish software to obtain an extensive bibliography of the academic literature on PRS. This tool collects and analyzes citation data from multiple sources, including Google Scholar, Microsoft Academic Search, PubMed, Scopus, ScienceDirect (Elsevier), ACM Digital Library, Springer Link, IEEE/IEE Library, and Francis. It provides various citation metrics, such as article counts, and total citations (*Harzing, 2010*).

Based on the plot of the number of publications on PRS topics in Fig. 1, there has been a steady growth in related PRS publications since 2013. Therefore, the first search focused on the time frame between 2013 and 2023, utilizing fundamental search parameters, including the phrases and search terms, such as "polygenic risk score" or "polygenic risk scores tool", "predictive polygenic risk score", and "polygenic risk score software". Most of the studies are found in the databases of PubMed, Google Scholar, Semantic Scholar, arXiv, ScienceDirect, and IEEE Xplore. The subsequent section demonstrates the criteria we have chosen.

### Selection criteria

Three criteria were established for the inclusion and subsequent analysis of PRS articles. Any articles failing to adhere to these criteria were omitted:

- The review ensured articles that cover methods, or software for generating polygenic risk scores.
- Articles must be relatively current. In this regard, we chose articles that were published between 2013 and 2023. This 10-year period could correspond to the main research period of interest for the PRS topics. Articles are required to be of recent publication. Consequently, we selected articles released within the timeframe of 2013 to 2023. This decade may align with the principal era of research significance for PRS subjects.
- Exclusion of book chapters, meeting abstracts, conference proceedings, workshop descriptions, non-English articles, and master and doctoral dissertations.

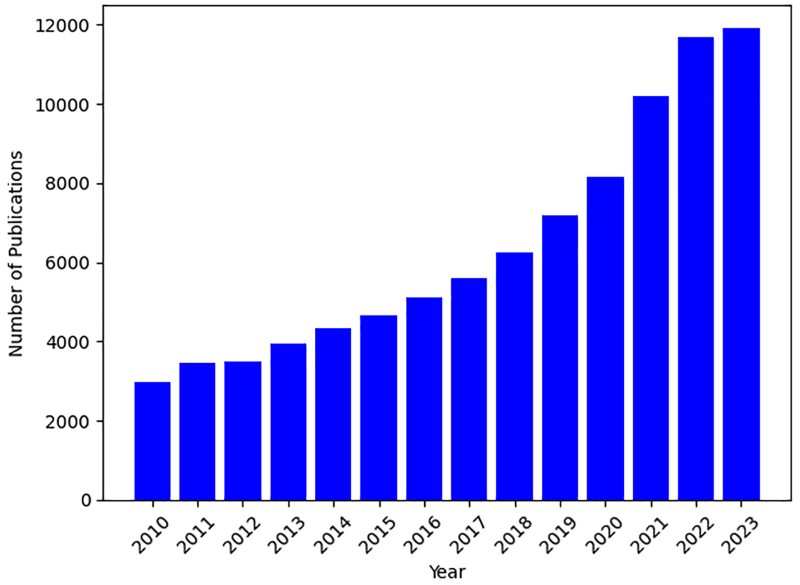

**Figure 1** **Temporal trends in PRS publications from 2010 to 2023.**

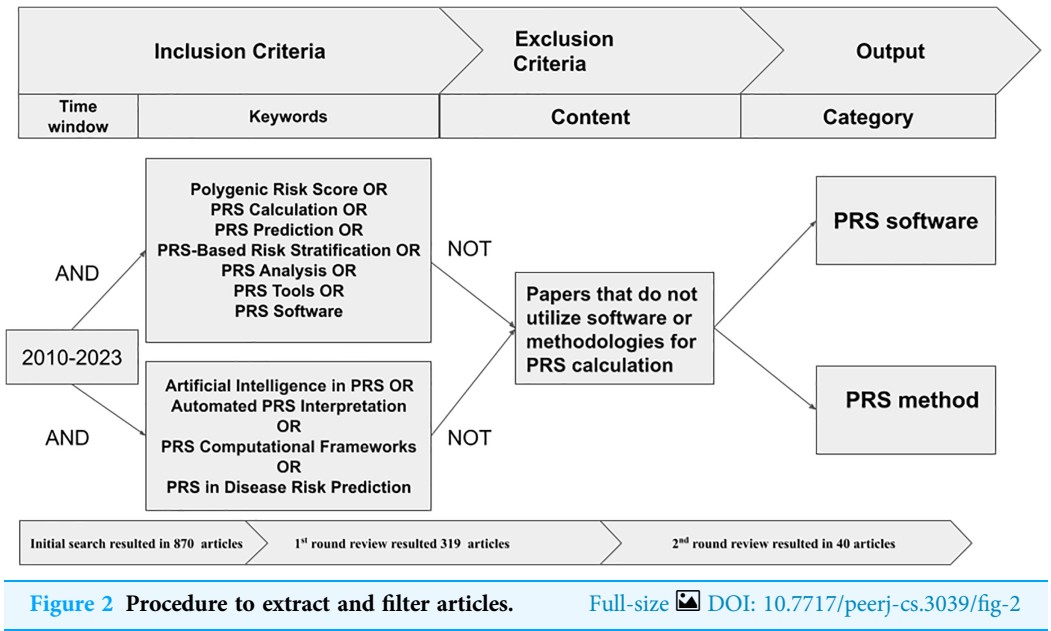

**Figure 2** **Procedure to extract and filter articles.**

## Filtering/reviewing process

The goal is to find articles that focus on the software and methods for generating PRS. We manually screened each article in three rounds and classified them.

We initially had 870 articles that matched the criteria. However, some of them were duplicates from different databases. We eliminated 91 duplicate articles and proceeded to the manual screening rounds. We only kept the articles that discussed the methods, tools,

or software for generating PRS. We then sorted them into categories. We conducted the review as following:

- **First round:** We reviewed the titles, abstracts, keywords, and conclusions of each article and discarded those that did not meet the selection criteria. This left us with 319 articles for the next round of review.
- **Second round:** The full texts of the remaining articles were reviewed to ensure they met the criteria, narrowing the selection to 40 articles for the final round. In this round, we conducted an in-depth analysis of each article, focusing on the main theme, and journal rank. Ultimately, we selected and analyzed the most relevant articles. Figure 2 shows the process of filtering and extracting academic articles from the initial search results.

## CLASSIFICATION METHOD

We categorized the literature on PRS by their research topics, selecting and filtering 40 articles. These were divided into two main groups: PRS software and PRS methods. The PRS methods were further subdivided into four categories: threshold-based methods, penalized regression methods, Bayesian methods, and machine learning methods. The PRS software was classified into three categories: Command-line, Web Application, and Library. Some methods have dedicated software, which are categorized under both PRS methods and PRS software.

## METHODS TO GENERATE PRS

The basic stepwise process for calculating PRS adds up the effects of many genetic variants that are linked to the trait or disease (*Chung, 2021*). Each variation has a weight that shows how much it influences the trait or disease. The formula to calculate the PRS for a person is:

$$\text{PRS}_j = \sum_{i=1}^{N} \beta_i * \text{dosage}_{ij}$$

where $N$ represents the count of SNPs in the score, $\beta_i$ is the effect size (or beta) of variant $i$ and dosage refers to the number of copies of SNP $i$ present in the genotype of individual $j$ (*Chung, 2021*).

The coefficients are usually derived from a large study that compares the genomes of people with and without the trait or disease (*Collister, Liu & Clifton, 2022*). Only the variants that have a meaningful effect on the trait or disease are selected for the score based on a statistical test ($p$-value cutoff). The main well-known methods found in the articles are cited in the following sub-sections.

### Threshold-based methods
#### C+T
The Clumping and Thresholding (C+T) method is a widely used approach for calculating PRS. It identifies genome-wide significant variants and groups them based on linkage disequilibrium (LD), excluding those in strong LD with an index variant that has the

lowest $p$-value in each group, this process helps to identify independent genetic variants associated with a trait (*Wray, Goddard & Visscher, 2007*; *Euesden, Lewis & O'reilly, 2015*). The method operates on the premise that only a few single nucleotide polymorphisms (SNPs) have non-zero effects on the trait. Genetic variants are first clumped based on LD and then filtered based on their $p$-values to derive polygenic scores (*Kim et al., 2023*; *Mak et al., 2017*).

Breast cancer polygenic risk scores for non-European populations, underrepresented in genetics studies, were developed using the C+T method (*Ho et al., 2022*). It was also used to compare different polygenic profiling methods for Alzheimer's disease risk (*Leonenko et al., 2021*). This method was utilized to assess the race-specific susceptibility of SNPs to AS in the Taiwanese population, as well as to examine the connection between SNPs associated and HLA-B27 with ankylosing spondylitis (AS) susceptibility (*Ko et al., 2022*). A meta-analysis used this method to determine the influence of PRS on the risk of coronary artery disease (*Agbaedeng et al., 2021*). A PRS for autoimmune Addison's disease was constructed and evaluated using the C+T method (*Aranda-Guillé et al., 2023*). Using the UK Biobank dataset, polygenic risk scores for elevated intraocular pressure, a risk factor for glaucoma, were constructed with the C+T method as described in *Gao, Huang & Kim (2019)*.

### SCT

The Stacked Clumping and Thresholding (SCT) is an extension of C+T that allows more flexibility in selecting SNPs based on four criteria: $p$-value threshold, LD window size, LD correlation threshold, and imputation accuracy (*Privé et al., 2019*). SCT generates PRSs for different settings of these criteria and then selects the optimal ones using a penalized regression approach on the validation data (*Privé et al., 2019*).

## Penalized regression methods

### LASSO

The Least Absolute Shrinkage and Selection Operator (LASSO) is a method used in regression analysis that performs both variable selection and regularization. The goal of LASSO is to achieve the smallest possible sum of squared errors, with the condition that the total absolute value of the coefficients does not exceed a predetermined threshold (*Ribbing et al., 2007*). It is used in machine learning and statistics to select variables in a model by shrinking some of the coefficients to zero. This method helps to prevent overfitting by decreasing the number of variables included in the model. LASSO can be used for PRS development. In this context, LASSO is used as a variable selection technique to select the most important genetic variants for inclusion in the PRSs. Addressing the insufficient representation of non-European communities in genetics studies to develop breast cancer polygenic risk scores utilizing the LASSO method as indicated in *Ho et al. (2022)*.

### Lassosum

Similar to LASSO, and it applies a LASSO penalty to nullify the effect sizes of genetic variants. It further prunes genetic variants exhibiting linkage disequilibrium and applies a

threshold to the remaining variants based on their _p_-values (_Mak et al., 2017_). The Lassosum approach was reported in several PRS studies. An assessment was made on the race-specific susceptibility of single nucleotide polymorphisms to ankylosing spondylitis (AS) in the Taiwanese population. The association between human leukocyte antigen (HLA)-B27 and susceptibility SNPs for AS in Taiwan was explored. Polygenic risk scores were used to analyze genetic variations in predicting the development of AS using LassoSum as indicated in _Ko et al. (2022)_. A meta-analysis investigated how PRSs affect the likelihood of developing coronary artery disease by employing the Lassosum method (_Agbaedeng et al., 2021_). A reference-standardized framework assessed the predictive value of several polygenic risk score methodologies, including Lassosum (_Pain et al., 2021_).

### SBLUP

The Super Genomic Best Linear Unbiased Prediction (SBLUP) technique adjusts SNP effect magnitudes by utilizing an external LD reference panel. This process transforms the ordinary least squares estimates of SNP into nearly optimal linear unbiased predictions (_Ren et al., 2021_). The SBLUP utilizes a Bayesian framework to calculate the magnitude of genetic variants effects. This method presumes that the distribution of these effects is normal, centering around zero, with their variance inversely proportional to the number of variants in the score. The SBLUP technique provides greater precision in the construction of PRS compared to other methods (_Slunecka et al., 2021_; _Robinson et al., 2017_).

## DBSLMM

The Deterministic Bayesian Sparse Linear Mixed Model (DBSLMM) is a technique used to compute polygenic risk scores. This method utilizes a versatile approach to modeling the distribution of effect sizes. This allows for strong and precise predictions over various genetic structures. Additionally, DBSLMM employs a straightforward deterministic search method to produce an estimated analytical solution based solely on summary statistics. Through simulation tests, DBSLMM has demonstrated its ability to provide scalable and precise predictions for a wide array of genuine genetic structures (_Yang & Zhou, 2020_).

### Bayesian methods

#### LDpred

The linkage disequilibrium pred (LDprep) method is widely utilized for the calculation of PRS. It operates by using summary statistics alongside a matrix that measures the correlation among genetic variants. It's a Bayesian approach that accounts for LD among genetic variants, assuming that each variant independently affects the trait. LDpred is a two-step method: it first estimates the LD structure from a reference panel and then uses this structure to adjust Genome-373 Wide Association Study (GWAS) summary statistics for the effects of LD. This method requires defining a tuning parameter ($\rho$), which is an estimate of the genetic variants assumed to be causal (_Imam, Noguera & Donohue, 2014_; _Vilhjalmsson et al., 2015_).

An evaluation was conducted on the race-specific SNP susceptibility of AS in Taiwanese people, as well as the relationship between HLA-B27 and AS susceptibility SNPs in Taiwan. A PRS technique was also used to examine genetic variations in predicting the

development of AS using LDpred (*Ko et al., 2022*). A meta-analysis investigated how PRSs affect the likelihood of developing coronary artery disease, utilizing the method described by *Agbaedeng et al. (2021)*. Additionally, LDpred was utilized to construct PRS to analyze the contribution of common genetic variations to suicide attempts. The aim was to demonstrate the genetic overlap and correlation between measures of suicide attempts and to explore the genetic associations of suicide attempts with other traits, such as insomnia and psychiatric disorders (*Ruderfer et al., 2020*).

### JAMPred

The Joint Analysis of Marginal Summary Statistics Prediction (JAMPred) is a technique for computing polygenic risk scores based on summary data from GWAS and a reference genotyping panel (*Newcombe et al., 2019*). JAMPred considers linkage disequilibrium among genetic variations and uses a Bayesian framework to estimate impact sizes and posterior probability of inclusion for each variant, furthermore, JAMPred employs variable selection and model averaging techniques to enhance the accuracy and stability of polygenic risk scores (*Newcombe et al., 2019*). An example of how PRS was utilized as a predictive tool for identifying high-risk patients with Parkinson's disease. Various methods, including JAMPred as described in *Shan et al. (2021)*.

### SBayesR

The SBayesR method is a Bayesian approach, which often used to compute polygenic risk scores, It incorporates a spike-and-slab technique to model the effects of genetic variants on the phenotype of interest (*Pham et al., 2022*). Efforts have been made to elucidate the most effective methodologies for polygenic profiling when screening individuals for Alzheimer's disease risk. Various methods, including the SBayesR approach, are employed utilizing datasets sourced from prominent institutions such as the UK Biobank and National Institute on Aging Genetics of Alzheimer's Disease Data Storage Site (NIAGADS) (*Leonenko et al., 2021*).

A reference-standardized framework was applied to assess the predictive value of several PRS methods. SBayesR was among the best methods found to assess the predictive value of several PRS as indicated in *Pain et al. (2021)*.

### EB-PRS

The emperical Bayes polygenic risk score (EB-PRS) is a method to generate PRS from summary statistics of GWAS and employs a statistical approach known as empirical Bayes to estimate the impact sizes of genetic markers throughout the whole genome, EB-PRS does not need parameter tuning or the use of external data (*Song et al., 2020*). The EB-PRS method has proven to be effective in producing outstanding outcomes independently, without the need for parameter tuning or external datasets. However, research indicates that its performance can be enhanced further when a reference panel is utilized (*Adam et al., 2022*).

### BridgePRS

The BridgePRS technique is a Bayesian polygenic risk score strategy that combines the PRS of two populations with differing ancestry. The goal is to address the PRS Portability

Problem by utilizing common genetic effects across ancestries to improve PRS accuracy in non-European communities. In other words, it seeks to increase the accuracy of PRS estimations for people from different and underrepresented heritage groups (*Hoggart et al., 2023*).

### Machine learning methods

The computation of PRS often relies on simple linear models which might not fully encompass the intricate interdependencies involved between phenotypes and genotypes (*DeWan, 2018*; *Aschard, 2016*). Therefore, machine learning methods that can account for non-linearities and interactions among genetic variants are of interest for improving the accuracy and interpretability of PRS.

A proposed approach combines an ensemble method for selecting SNPs with Gradient-Boosted Trees (GBT) to account for the non-linear and interactive influences of SNPs on phenotypes. When a PRS is included as a feature within an extreme gradient boosting model, there is a notable enhancement in the explained variance percentage relative to the conventional linear PRS model, as observed across nine complex phenotypes within a diverse ancestral group from the UK Biobank (*Elgart et al., 2022*).

The utilization of machine learning and deep learning techniques was thoroughly explored to compute polygenic risk scores from GWAS data. Random forest (RF) and support vector machines (SVM), and deep learning methods were employed to calculate weight vectors, which play a pivotal role in PRS computation (*Öztornaci et al., 2023*). Additionally, variable importance measurements obtained from the RF method serve as weight vectors. In all these methods, individual risk scores are derived by multiplying each SNP with its corresponding weight vector.

*Peng et al. (2024)* introduced a DL framework that captures intricate genetic interactions beyond additive effects. In contrast to traditional PRS models, which often assume linear relationships, DeepRisk leverages neural networks to model non-linear associations among single-nucleotide polymorphisms (SNPs). The approach described in *Peng et al. (2024)* has demonstrated superior performance in predicting disease risk, particularly in scenarios involving complex genetic architectures. The study (*Zhou et al., 2023*) introduced a neural network model that captures non-linear interactions among genetic variants, offering enhanced predictive accuracy and deeper insights into disease mechanisms. The PRS-Net, as described in *Li et al. (2024)*, demonstrated that incorporating a lightweight geometric layer into gene-level PRSs yields reproducible and biologically interpretable improvements over both linear and black-box non-linear baselines, particularly for immune-mediated diseases and heterogeneous ancestries.

## RESULTS

We extracted a collection of articles on polygenic risk scores (PRS) from online databases. Each article was thoroughly reviewed and classified according to the established categorization method. Figure 3 presents the PRISMA flow diagram outlining the selection process.

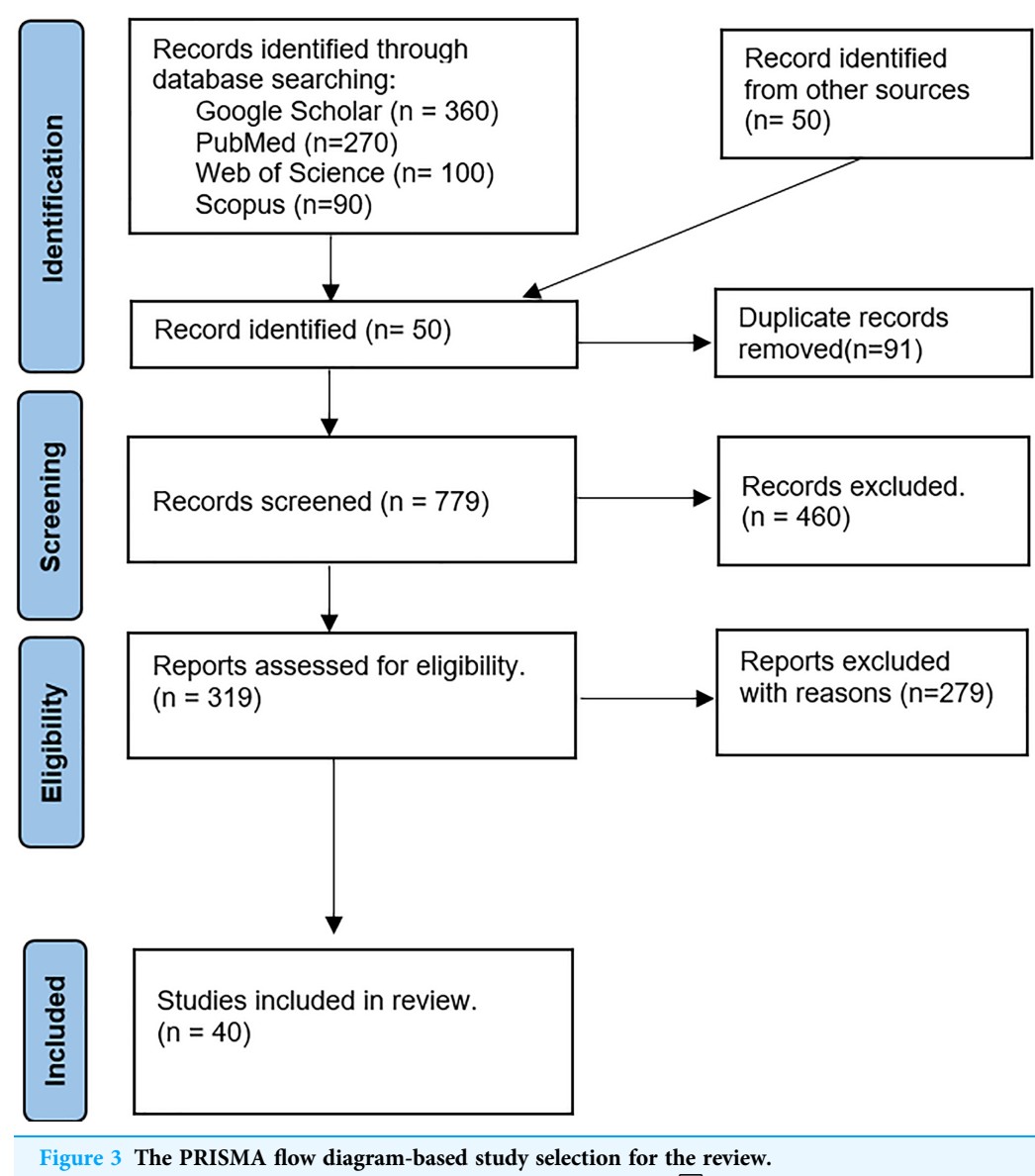

**Figure 3** **The PRISMA flow diagram-based study selection for the review.**

Table 1 highlights a range of software tools commonly utilized in PRS analysis, grouped into three main types: command-line tools, web-based applications, and programming libraries. The table also outlines the diverse analytical methods employed in PRS studies.

## DISCUSSION

### The benefits and limitations of the techniques used
#### *Threshold-based methods*

These methods use a predefined threshold to identify variants by utilizing their $p$-values or the magnitude of their effects derived from GWAS summary data. Research has suggested that threshold-based methods are simple and relatively easy to implement, and widely used

**Table 1 Summary of PRS software and methods.**

| Category | Software/ method | Programming language | Availability | Description | Ref |
|---|---|---|---|---|---|
| Command-line | Plink2 | C++ | Free | GWAS analysis and research in population genetics | *Chang et al. (2015)* |
| | PRSice | C++, R | Free | Computing, implementing, assessing, and graphically representing PRS results with R | *Euesden, Lewis & O'reilly (2015)* |
| | PRSice2 | C++, R | Free | Automating and simplifying the analysis of PRS on extensive datasets | *Choi & O'Reilly (2019)* |
| | PRS-CS | Python | Free | Infers posterior SNP effects using continuous shrinkage priors and LD panels | *Ge et al. (2019)* |
| | PRS-on-Spark (PRSoS) | Python, Spark | Free | Computes PRS handling various inputs and ambiguous SNPs | *Chen et al. (2018)* |
| | EraSOR | Python | Free | Eliminates bias from overlapping samples in GWAS/PRS data | *Choi et al. (2023)* |
| | BridgePRS | R, Python | Free | Bridges PRS across populations to address portability issues | *Hoggart et al. (2023)* |
| | AnnoPred | Python | Free | Predicts disease risk integrating GWAS statistics and annotations | *Hu et al. (2017)* |
| Web application | Cancer PRSweb | – | Free | Online repository hosting PRS for major cancer traits | *Fritsche et al. (2020)* |
| | CanRisk | – | Free | Estimates breast/ovarian cancer risks and mutation probabilities | *Carver et al. (2021)* |
| Library | bigsnpr | R | Free | Calculates PRS using GWAS statistics; supports LDpred2 | *Privé, Arbel & Vilhjálmsson (2020)* |
| | EB-PRS | R | Free | Uses effect size distribution without tuning or external data | *Song et al. (2020)* |
| | Lassosum | R | Free | Penalized regression on GWAS summary statistics *via* Lasso | *Mak et al. (2017)* |
| | PolyFun | Python | Free | Fine-mapping and prediction including PolyFun, PolyLoc, PolyPred | *Weissbrod et al. (2022)* |
| | XPXP | Python | Free | Enhances PRS prediction using cross-population/phenotype analysis | *Xiao et al. (2022)* |
| | LDpred | Python | Free | Estimates posterior effect sizes using LD and prior models | *Vilhjalmsson et al. (2015)* |
| Threshold-based | C+T | – | – | Selects SNPs by *p*-values and LD, sums risk alleles | *Ho et al. (2022)*, *Leonenko et al. (2021)*, *Ko et al. (2022)*, *Agbaedeng et al. (2021)*, *Aranda-Guillé et al. (2023)*, *Gao, Huang & Kim (2019)* |
| | SCT | – | – | Combines multiple C+T scores with stacking classifier | *Privé et al. (2019)* |
| Penalized regression | Lasso | R (Lassosum) | – | Selects SNPs and estimates effects *via* Lasso | *Ho et al. (2022)* |

*(Continued)*

| Category | Software/method | Programming language | Availability | Description | Ref |
|---|---|---|---|---|---|
| | Lassosum | R | – | Lasso penalty on GWAS statistics; handles overfitting | *Ko et al., (2022)*, *Agbaedeng et al., (2021)*, *Pain et al. (2021)* |
| | SBLUP | – | – | Bayesian method accounting for LD and SNP interactions | *Robinson et al. (2017)*, *Slunecka et al. (2021)* |
| | DBSLMM | – | – | Sparse mixed model approximation using heritability tuning | *Yang & Zhou, (2020)* |
| Bayesian | LDpred | Python | Free | Gibbs sampling with LD to estimate PRS | *Imam, Noguera & Donohue (2014)*, *Vilhjalmsson et al. (2015)*, *Ko et al. (2022)*, *Agbaedeng et al. (2021)* |
| | Jampred | – | – | Joint analysis across GWAS to improve power and accuracy | *Newcombe et al. (2019)*, *Shan et al. (2021)* |
| | EB-PRS | R | Free | Uses effect size distribution without external info | *Song et al. (2020)* |
| | SBayesR | – | – | Bayesian regression with spike-and-slab prior | *Pham et al. (2022)*, *Leonenko et al. (2021)*, *Pain et al. (2021)* |
| | BridgePRS | R, Python | Free | Combines Bayesian PRS from distinct ancestries | *Hoggart et al. (2023)* |
| Machine learning | RF | – | – | Uses random forest to weight SNPs for PRS | *Öztornaci et al. (2023)* |
| | SVM | – | – | Uses support vector machines to improve classification | *Öztornaci et al. (2023)* |
| | GBT | – | – | Gradient boosting trees + XGBoost for non-linear SNP effects | *Elgart et al. (2022)* |

in practice (*Privé et al., 2019*; *Lewis & Vassos, 2020*). They can provide a straight forward interpretation of the PRS as a weighted sum of selected variants. However, research has shown that threshold-based methods may ignore variants with small effects that can collectively contribute to the PRS, they are sensitive to the choice of threshold, which can affect the predictive performance and the number of variants included (*Lewis & Vassos, 2020*).

### Penalized regression methods

These methods use a regularization term to penalize the coefficients of the PRS, which can handle high-dimensional data effectively, they have been shown to reduce overfitting by shrinking coefficients, especially for correlated or weakly associated variants (*Privé, Aschard & Blum, 2019*). They also provide variable selection by shrinking less informative variables towards zero, which can improve interpretability and reduce collinearity. However, the choice of penalty parameters can be subjective and depend on the data and the trait (*Pattee & Pan, 2020*). The interpretation of the resulting coefficients may be challenging, especially when using non-linear penalties or complex models, they also assume a linear relationship between predictors and outcomes, which may not hold for some traits or diseases (*Pattee & Pan, 2020*).

### Bayesian techniques

These methods use a probabilistic framework to estimate the posterior distribution of the PRS given the prior information and the data. They can incorporate various sources of information such as functional annotations and biological pathways. Also they account for uncertainty and provide credible intervals for the PRS (*Ge et al., 2019*; *Zhou, Qie & Zhao, 2023*). They have the flexibility to incorporate prior knowledge and choose the prior distribution (*Ge et al., 2019*). However, they can be computationally intensive, they may also require expertise in Bayesian statistics for proper implementation and interpretation of results (*Song et al., 2020*).

### Machine learning methods

These methods use various algorithms to learn the optimal PRS from the data, such as random forests, support vector machines, neural networks, *etc* (*Öztornaci et al., 2023*). Studies have shown that they can capture complex and nonlinear relationships between variants and outcomes, which linear models may not capture, they optimize predictive performance by using cross-validation, grid search, or other techniques to tune the hyperparameters, and have flexibility in feature engineering, such as using interactions, transformations, or embeddings of variants (*Squires, Weedon & Oram, 2023*; *Mamani, 2020*). However, they may overfit the data if not properly regularized, which can reduce generalizability and robustness. Also, they can be challenging to interpret, particularly in the case of intricate models such as neural networks, which are often perceived as black boxes. They have been shown to have high computational complexity, which can limit their scalability and applicability (*Mamani, 2020*).

## The applications of PRS

Polygenic risk scores are a versatile tool for healthcare applications (*Slunecka et al., 2021*). PRS can estimate the probability of an individual having or developing a specific disease or trait, facilitating risk stratification and early intervention that consider both genetic and environmental factors (*Lewis & Green, 2021*; *Corpas & Fatumo, 2023*). These interventions can range from lifestyle changes to preventive surgeries, depending on the condition and the individual's preferences (*Lewis & Green, 2021*). PRS can provide healthcare providers with valuable information for risk assessment, disease management, and preventive care, and help them deliver tailored advice and interventions to their patients (*Chapman, 2023*). Additionally, PRS can inform the design of screening programs and research studies, by dividing populations into different risk groups and adjusting screening criteria and intervention effects accordingly (*Slunecka et al., 2021*). However, the implementation of PRS-based approaches in healthcare requires careful attention to ethical, social, and practical issues, to ensure fair and respectful practices that protect patient autonomy and privacy while maximizing benefits for individuals and society (*Chapman, 2023*; *Aragam & Natarajan, 2020*; *Lewis & Vassos, 2020*).

## Challenges and future directions

### Increasing diversity and representation of data

PRS methods predominantly rely on data from individuals of European ancestry, leading to limitations in their applicability and generalizability to other ethnic groups. This lack of diversity can result in biased risk predictions and may exacerbate existing health disparities. There is a growing recognition of the importance of including diverse and representative genetic data from different populations. Initiatives are underway to collect and analyze genomic information from historically underrepresented populations, such as the Human Heredity and Health Africa (H3Africa) initiative. Developing methods that account for genetic diversity and mixed ancestry within varied groups is essential for improving the precision and reliability of PRS across all ethnic backgrounds (*Zhang et al., 2023*; *Lam et al., 2019*).

To enhance multi-ancestry prediction models, it is necessary to leverage genetic data from diverse populations to improve PRS performance. For example, the PROSPER method demonstrates improved precision and reliability showing a 70% increase in accuracy for individuals of African ancestry compared to traditional models (*Zhang et al., 2024*). Such approaches help correct biases inherent in European-centric models by accounting for differences in genetic architecture, including linkage disequilibrium patterns and allele frequencies (*Cavazos & Witte, 2021*). Incorporating ancestry-specific data into PRS algorithms is essential for improving their applicability across diverse populations (*Lerga-Jaso et al., 2024*).

A major driver of improved accuracy and generalizability in PRS development is the increasing availability of large-scale biobank databases. Due to the complexity of the human genome, large datasets are critical for identifying associations between genetic variants and complex traits (*Raben et al., 2023*). Biobank resources support the development, validation, and application of PRS by providing extensive training data and, crucially, multi-ancestry samples for cross-population evaluation (*Tsuo et al., 2024*; *Thompson et al., 2022*). Major global initiatives including the UK Biobank, All of Us (AoU), China Kadoorie Biobank, Biobank Japan, deCODE Genetics, the Estonian Biobank, and Lifelines in the Netherlands are helping to address the historical underrepresentation of non-European populations in genetic research (*Ju et al., 2022*). These resources provide the statistical power needed to reduce false positives, identify novel variants, and refine estimates of single nucleotide polymorphism (SNP) effect sizes (*Raben et al., 2023*; *Ju et al., 2022*). With advancements in analytic tools and machine learning algorithms, biobank databases are making PRS construction increasingly accessible to a broader range of researchers (*Sakaue et al., 2020*; *Du et al., 2023*).

### Using explainable AI for improving the interpretability of PRS

ML-based methods can be used to learn the optimal PRS from the data. However, the lack of transparency of ML's prediction could lead to a poor generalization on datasets when a model learns to predict on irrelevant features. The explainability of MLs is crucial in healthcare as the consequences of a wrong prediction in diagnostics may cause life-changing decisions for a patient (*Elgart et al., 2022*). Recently, explainable artificial

intelligence (XAI) has been widely used in the literature to overcome the lack of insight of ML-based models in healthcare and medical diagnosis systems (*Zhang, Weng & Lund, 2022*). XAI reveals the decision patterns of ML-based models, which helps medical practitioners understand the logical reasoning for the model's prediction (*Zhang, Weng & Lund, 2022*). XAI is a promising research direction that requires more attention from the PRS research community.

### Incorporating environmental and lifestyle factors

PRS methods typically focus solely on genetic components and may not fully capture the multifactorial nature of complex traits and diseases. However, lifestyle and environmental factors—such as exercise, diet, and exposure to toxins—play significant roles in disease risk but are often overlooked in traditional PRS analyses. Integrating these non-genetic variables into PRS models can enhance their predictive power and relevance. For instance, including data on smoking behavior, socioeconomic status, and geographic location can improve risk stratification and facilitate more personalized interventions. Advanced statistical approaches, such as gene-environment interaction modeling, are being developed to better capture the interplay between genetic and environmental influences on health outcomes (*Wang et al., 2021*; *Koch et al., 2023*).

Recent advances demonstrate the potential of integrating such factors. For example, incorporating 109 exposome variables—including tobacco use, education, and others—into cardiovascular disease risk prediction using machine learning increased the area under the curve (AUC) to 0.82 (*Shahbazi & Nowaczyk, 2025*). Additionally, the use of Internet of Things (IoT) devices enables real-time integration of lifestyle and environmental data such as diet and air quality, supporting adaptive health interventions for hyper-personalized medicine (*Tan et al., 2025*). Moreover, social determinants of health (SDoH) have shown significant associations with disease outcomes, particularly in high-risk environments, further highlighting the importance of integrating socio-environmental context into PRS applications (*Guare et al., 2024*).

### Evaluating clinical and public health implications

PRSs show great potential for predicting health risks, but their implementation also raises important ethical, societal, and legal concerns. The integration of PRS into clinical practice requires a thoughtful approach to issues such as informed consent, data privacy, protection, and equitable access to genetic services and interventions. It is essential to rigorously evaluate both the effectiveness and broader implications of PRS across diverse populations and clinical settings. Studies assessing the utility of PRS in guiding preventive strategies, screening programs, and therapeutic decisions are vital for shaping evidence-based healthcare policies.

Public understanding and health literacy around genetic risk are equally important. Enhancing awareness can help ensure informed decision making and reduce the risk of misinterpretation or misuse of genetic information in clinical and personal contexts.

Ethical considerations must be addressed, particularly when individuals are assigned high-risk scores, which may lead to psychological distress or stigmatization. In addition,

disparities in access to genetic testing, risk assessment tools, and personalized interventions can exacerbate existing health inequalities (*Andreoli et al., 2024*). Clear communication of genetic findings and proper implementation supported by robust informed consent processes are essential for integrating PRS into personalized medicine and improving shared decision-making between patients and healthcare providers (*King & Bishop, 2017*).

In summary, addressing these challenges and leveraging the opportunities presented by advancements in genomics, data science, and healthcare delivery will be essential for realizing the complete promise of PRS in enhancing health outcomes and minimizing health inequities among different groups (*Koch et al., 2023*; *Simona et al., 2023*; *Lewis & Vassos, 2017*). The criticality of generating precise PRS for various complex traits and illnesses is paramount. PRS offers a gauge of an individual's genetic susceptibility to a complex trait or illness, indicating the probability of manifesting a specific trait or illness grounded on one's genetic makeup. PRS analysis aims to pinpoint individuals at heightened disease risk by analyzing genetic variations alongside clinical factors. Thus, the more precise the PRS, the more effectively we can pinpoint disease risks and devise preventative measures.

## CONCLUSION

Polygenic risk scores hold immense promise for predicting disease. The extensive literature on PRS research reflects the growing interest and investment in this field, with significant advancements in methods, tools, and applications. By integrating genetic information with clinical data, PRSs contribute to predicting disease risk and guiding preventive interventions. However, challenges remain in ensuring data diversity, incorporating environmental factors, and addressing ethical considerations. Future research efforts should focus on overcoming these challenges to unlock the full potential of PRSs in improving clinical outcomes and public health interventions.

## LIST OF ABBREVIATIONS

| | |
|---|---|
| **PRSs** | Polygenic risk scores |
| **PRS** | Polygenic risk score |
| **ML** | Machine learning |
| **PRISMA** | Preferred Reporting Items for Systematic Reviews and Meta-Analyses |
| **SNPs** | Single Nucleotide Polymorphisms |
| **C+T** | Clumping and Thresholding |
| **LD** | Linkage Disequilibrium |
| **AS** | Ankylosing Spondylitis |
| **HLA** | Human Leukocyte Antigen |
| **SCT** | Stacked Clumping and Thresholding |
| **Lasso** | Least Absolute Shrinkage Selection Operator |
| **SBLUP** | Super Genomic Best Linear Unbiased Prediction |
| **DBSLMM** | Deterministic Bayesian Sparse Linear Mixed Model |
| **LDpred** | Linkage Disequilibrium pred |

| | |
|---|---|
| **GWAS** | Genome-373 Wide Association Study |
| **JAMPred** | Joint Analysis of Marginal Summary Statistics Prediction |
| **EB-PRS** | Empirical Bayes-PRS |
| **SVM** | Support vector machines |
| **RF** | Random Forests |
| **GBT** | Gradient-Boosted Trees |
| **CS** | Continuous Shrinkage |
| **H3Africa** | Human Heredity and Health in Africa |
| **XAI** | Explainable AI |

### Funding
The research received financial support from the King Salman Center for Disability Research under the Research Group number KSRG-2023-438. The funders had no role in study design, data collection and analysis, decision to publish, or preparation of the manuscript.

### Grant Disclosures
The following grant information was disclosed by the authors:
King Salman Center for Disability Research: KSRG-2023-438.

### Competing Interests
The authors declare that they have no competing interests.

### Author Contributions
- Sara Benoumhani conceived and designed the experiments, performed the experiments, analyzed the data, performed the computation work, prepared figures and/or tables, authored or reviewed drafts of the article, and approved the final draft.
- Areej Al-Wabil performed the experiments, analyzed the data, performed the computation work, authored or reviewed drafts of the article, and approved the final draft.
- Niddal Imam performed the computation work, authored or reviewed drafts of the article, and approved the final draft.
- Bashayer Alfawaz performed the computation work, authored or reviewed drafts of the article, and approved the final draft.
- Amaan Zubairi performed the computation work, authored or reviewed drafts of the article, and approved the final draft.
- Dalal Aldossary performed the computation work, authored or reviewed drafts of the article, and approved the final draft.
- Mariam AlEissa conceived and designed the experiments, performed the experiments, analyzed the data, performed the computation work, authored or reviewed drafts of the article, and approved the final draft.

## Data Availability

This is a literature review.

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
