# Peer review of "A review of methods and software for polygenic risk score analysis"

_PeerJ Computer Science, doi:10.7717/peerj-cs.3039_

## Round 0.1 · original submission · Major Revisions

The reviewers raised s number of issues that need to be addressed, therefore the article is unsuitable for publication in its current form.

Please prepare an enhanced version of the manuscript taking into account the suggestions and the comments provided by the reviewers.

Reviewer 1 ·

Basic reporting

This review article explores the advancements in PRS research, discussing methodologies, software tools, and applications across various disciplines. It systematically reviews the literature, identifying 60 relevant articles classified based on PRS methods and software. The article introduces key PRS calculation methods, including penalized regression and threshold-based approaches, and discusses Bayesian and machine learning methods, as well as prominent software and their features. It highlights the applications of PRS in disease prevention, along with the challenges and future directions. Overall, this is a well-conducted review that helps readers quickly grasp the field. However, there are a few minor issues that need revision and clarification.

Experimental design

There are no major issues with the method of literature search. However, the authors repeatedly emphasize that 60 relevant articles were reviewed. Neither the total number of articles mentioned in the paper (80 articles) nor the tables in the text (which seem to include only 40 articles) clearly indicate which articles are part of the described 60. It would be helpful to include a table summarizing these 60 articles, highlighting their significance for readers.

Validity of the findings

1. Could the authors explain why the number of published articles in 2023 shows a decline in Figure 1?
2. The resolution of Figure 2 appears to be insufficient.
3. Figures 2 and 3 show a high degree of similarity, making it unclear to readers why the authors chose to present them as separate figures. Could the authors clarify the purpose of dividing these into two figures?

Additional comments

In the discussion, the authors provide a solid comparison of the strengths and weaknesses of the methodologies, helping readers to understand the differences between these approaches. The explanation of related applications is also well-articulated. However, the section on ancestry differences appears somewhat insufficient.

First, references 40 and 71 seem unrelated to ancestry-specific PRS. The authors should correctly cite relevant articles. More appropriate references for this section might include:

"Zhang, H., Zhan, J., Jin, J. et al. A new method for multiancestry polygenic prediction improves performance across diverse populations. Nat Genet 55, 1757–1768 (2023)."
"Lam M, et al. Comparative genetic architectures of schizophrenia in East Asian and European populations. Nat Genet. 2019 Dec;51(12):1670-1678. doi: 10.1038/s41588-019-0512-x. Epub 2019 Nov 18. PMID: 31740837; PMCID: PMC6885121."

The authors should also discuss the implications of multiancestry PRS and acknowledge the growing trend of PRS models being adjusted for multiancestry populations. This is becoming increasingly relevant in addressing disparities in prediction performance across diverse populations.

Cite this review as

Reviewer 2 ·

Basic reporting

This manuscript provides a comprehensive review of polygenic risk score (PRS) methods, discussing advances in computational approaches, software tools, and applications. The inclusion of future directions, such as the need for more diverse genetic data and the integration of environmental factors, aligns with ongoing challenges in the field. The topic is relevant and timely, addressing critical issues in personalized medicine and genetic prediction. However, there are significant areas that require attention and improvement before the paper can be published.
While the systematic review is well-structured and follows PRISMA guidelines, the methodology lacks sufficient detail regarding inclusion criteria and screening processes. For example, while 60 articles were selected, the justification for the final sample is not clear. This omission raises questions about the comprehensiveness of the review and its representation of the field. Furthermore, while the article categorizes PRS methodologies and software into well-defined groups, it often fails to critically assess the relative strengths and limitations of these methods in specific contexts, limiting the reader’s ability to discern actionable insights.
The discussion of challenges and future directions is commendable but remains superficial in certain areas. For example, the mention of integrating environmental and lifestyle factors into PRS models is an important observation, but the authors do not provide specific examples or strategies for achieving this. Similarly, the exploration of ethical considerations and the implications of PRS in clinical practice is underdeveloped, leaving a critical gap in the manuscript.
From a presentation standpoint, the manuscript is well-organized but overly detailed in sections discussing basic PRS computation methods, which are widely known in the field. The balance between foundational information and novel insights needs refinement. Additionally, some figures and tables, while visually appealing, do not add significant value to the narrative and could be streamlined to improve focus.

Experimental design

See 1.

Validity of the findings

See 1.

Additional comments

See 1.

Cite this review as

Reviewer 3 ·

Basic reporting

No comment.

Experimental design

No comment.

Validity of the findings

No comment.

Additional comments

Polygenic risk scores (PRS) are a crucial tool for understanding the genetic underpinnings of complex traits and diseases. This manuscript provides a comprehensive review of PRS methods, covering developments from 2013 to 2023 and suggesting future directions. Below are specific suggestions to further enhance the quality of the manuscript:

1. The title, "A Review of Advances, Applications, and Future Directions in Polygenic Risk Score Methods," includes the term "Applications," but the corresponding section is considerably shorter compared to other aspects of the manuscript. Expanding on the practical applications of PRS would add valuable depth and make the review more balanced.
2. Recent advancements in deep-learning-based methods for PRS estimation should be included in this review, such as PRS-Net[1], DeepRisk [2], and NN [3]. Studies have demonstrated that deep learning techniques outperform traditional PRS methods in certain contexts, and their inclusion would make the review more up-to-date and comprehensive.
3. The emergence of large biobank databases, such as the UK Biobank and FinnGen, has significantly influenced PRS research. A discussion on how these databases will facilitate the development of more accurate and generalizable PRS models would be highly beneficial.
4. Incorporating a timeline or visual representation of the evolution of various PRS methods would provide readers with a more intuitive understanding of the field’s progress over time. This would be a valuable contribution to the manuscript and help readers contextualize the advancements more effectively.

References:
[1] Li, Han, et al. "PRS-Net: Interpretable polygenic risk scores via geometric learning." International Conference on Research in Computational Molecular Biology. Cham: Springer Nature Switzerland, 2024.
[2] Peng, Jiajie, et al. "DeepRisk: A deep learning approach for genome-wide assessment of common disease risk." Fundamental Research 4.4 (2024): 752-760.
[3] Zhou, Xiaopu, et al. "Deep learning-based polygenic risk analysis for Alzheimer’s disease prediction." Communications Medicine 3.1 (2023): 49.

Cite this review as

---

## Round 0.2 · Minor Revisions

The authors significantly improved the article and the reviewers noticed it. However, some drawbacks are still present in the study, as the first reviewer noted, and therefore the manuscript cannot be accepted for publication in its current form. I invite the authors to address the remaining issues and to submit a new, improved version of the article.

Reviewer 1 ·

Basic reporting

The authors have substantially improved the clarity and readability of the manuscript. They corrected the previously inconsistent reporting of the number of reviewed articles and revised the narrative to better distinguish between the total number of articles discussed and those systematically reviewed. This point has been adequately addressed, and I have no further concerns regarding basic reporting.

Experimental design

The authors revised the methodology section to include more explicit inclusion and exclusion criteria and provided an updated flow diagram (Figure 2). They also addressed the issue of a decline in publications in 2023 by updating the dataset with more complete and recent records. This correction was appropriately handled and improves the credibility of the reported trend.

Validity of the findings

While the authors responded to earlier concerns about figure resolution and the similarity between Figures 2 and 3, and the updated versions show noticeable improvement, I still find that the image quality—especially upon magnification—may not yet meet publication standards. I suggest the authors ensure that both figures meet high-resolution requirements for print and digital formats.
In addition, the revision of the ancestry-related section is appreciated and strengthens the discussion of cross-population PRS applicability. However, I noticed that the reference cited in line 364 on page 19 (Reference 24) appears to focus specifically on glaucoma PRS in the UK Biobank, and may not be the most appropriate citation for the statement regarding global initiatives aimed at addressing ancestry imbalance in genetic research. A more suitable citation that directly discusses biobank efforts in increasing non-European representation should be used.

Additional comments

This is a timely and informative review on PRS methodologies. The authors have made substantial improvements in response to reviewer feedback, including figure updates, method clarification, and discussion enrichment. Nevertheless, a few final issues remain: ensuring figure resolution meets publication quality, clarifying the reference used in the ancestry discussion, and refining citation accuracy. I also recommend adding a brief mention of the following recent work from Taiwan:

Sun et al., 2024. Utility of polygenic scores across diverse diseases in a hospital cohort for predictive modeling. Nature Communications 15 (1), 3168.
This study demonstrates real-world PRS validation in an East Asian population using existing European-derived scores and would provide relevant context to the manuscript's discussion on cross-ancestry PRS generalizability.

Cite this review as

Reviewer 3 ·

Basic reporting

NA

Experimental design

NA

Validity of the findings

NA

Additional comments

Please also include a discussion of PRS-Net [1-2] in the Machine Learning Methods section. PRS-Net is a graph neural network-based method that has been demonstrated to achieve superior disease risk prediction compared to previously proposed PRS methods. I have no further comments and recommend that this work be published.

References
[1] Li H, Zeng J, Snyder M P, et al. Modeling gene interactions in polygenic prediction via geometric deep learning[J]. Genome Research, 2025, 35(1): 178-187.
[2] Li H, Zeng J, Snyder M P, et al. PRS-Net: Interpretable polygenic risk scores via geometric learning[C]//International Conference on Research in Computational Molecular Biology. Cham: Springer Nature Switzerland, 2024: 377-380.

Cite this review as

---

## Round 0.3 · accepted · Accept

The authors correctly addressed the points raised by the reviewers and therefore I can recommend this article for acceptance.